# Buffalo bbu-miR-493-5p Promotes Myoblast Proliferation and Differentiation

**DOI:** 10.3390/ani14040533

**Published:** 2024-02-06

**Authors:** Liyin Zhang, Dandan Zhong, Chengxuan Yao, Qingyou Liu, Deshun Shi, Mingsheng Jiang, Jian Wang, Zhaocheng Xiong, Hui Li

**Affiliations:** 1State Key Laboratory for Conservation and Utilization of Subtropical Agro-Bioresources, College of Animal Science and Technology, Guangxi University, Nanning 530004, China; zhangliyin9904@126.com (L.Z.); dany08@126.com (D.Z.); yaochengxuan007@163.com (C.Y.); ardsshi@gxu.edu.cn (D.S.); 13878896387@163.com (M.J.); wangjsci@126.com (J.W.); 2Guangdong Provincial Key Laboratory of Animal Molecular Design and Precise Breeding, School of Life Science and Engineering, Foshan University, Foshan 528225, China; qyliu-gene@fosu.edu.cn; 3Research & Development Affairs Office, Guangxi University, Nanning 530004, China

**Keywords:** buffalo, miRNA, muscle development, bbu-miR-493-5p, myoblast

## Abstract

**Simple Summary:**

The nutritional and economic value of buffalo meat is being recognized more and more, but the quality and yield of buffalo meat still need to be improved. MiRNAs play important roles in skeletal muscle development, but their regulatory role in buffalo muscle is unclear. Therefore, this study investigated the effect of miRNAs on the proliferation and differentiation of buffalo myoblasts and discussed their relationship with inflammation in muscle injuries. The results suggest that bbu-miR-493-5p, which is differentially expressed during the proliferative and differentiation stages of buffalo myoblasts, can regulate buffalo skeletal muscle development and may act as an anti-inflammatory agent for the treatment of inflammatory responses to muscle injury. These findings provide a basis for the future use of miRNAs for buffalo selection and improved animal health.

**Abstract:**

In recent years, the meat and dairy value of buffaloes has become a major concern in buffalo breeding, and the improvement of buffalo beef quality is key to protecting buffalo germplasm resources and solving the problem of beef supply. MiRNAs play a significant role in regulating muscle development. However, the precise mechanism by which they regulate the development of buffalo skeletal muscles remains largely unexplored. In this study, we examined miRNA expression profiles in buffalo myoblasts during the proliferation and differentiation stages. A total of 177 differentially expressed miRNAs were identified, out of which 88 were up-regulated and 89 down-regulated. We focused on a novel miRNA, named bbu-miR-493-5p, that was significantly differentially expressed during the proliferation and differentiation of buffalo myoblasts and highly expressed in muscle tissues. The RNA-FISH results showed that bbu-miR-493-5p was primarily located in the cytoplasm to encourage buffalo myoblasts’ proliferation and differentiation. In conclusion, our study lays the groundwork for future research into the regulatory role of miRNAs in the growth of buffalo muscle.

## 1. Introduction

Buffaloes are an indispensable livestock resource in developing countries, originally used for agriculture, and now they have become one of the various meat sources for human consumption [1]. Studies show that buffalo meat quality is no different from ordinary beef [2] and that buffaloes have great meat development value. Therefore, the development and use of molecular technology to breed high-quality buffaloes with fast growth, large meat yield, and high quality can not only effectively improve buffalo varieties and protect buffalo germplasm resources but also improve the quality and meat yield of buffalo beef and solve the problems of beef supply.

Skeletal muscle development is an important part of ananimal’s growth and development process. The various stages of muscle development directly affect growth rate, meat yield, and meat quality [3]. The development and maturation of skeletal muscle, including skeletal muscle precursor cell establishment, proliferation, myoblast fusion, differentiation, and functional formation, is a complex biological process regulated by multiple genes and non-coding RNAs [4]. This process is precisely regulated by a number of muscle-specific transcription factors and cell signaling pathways. In particular, the family of myogenic regulatory factors (MRFs) mediates the generation of skeletal muscle progenitor cells, of which MyoD and Myf5 are involved in the establishment of muscle progenitor cell populations, while MRF4 and myogenin promote the development of muscle progenitor cells towards differentiation [5,6]. Additionally, the MEF2 family synergizes with myogenin to initiate terminal differentiation [7]. Then, the balance between growth factor signaling pathways such as IGF and FGF and the myogenic inhibitory factor myostatin regulates myoblast proliferation and differentiation [8]. In addition, myogenic fiber production requires the expression of structural proteins such as actin and myosin [8], and genes such as Pax7 maintain the muscle stem cell population to ensure the regenerative capacity of muscle tissue [9].

The widespread involvement of miRNAs in the regulation of skeletal muscle development has been clearly demonstrated by numerous studies [10,11]. Non-coding RNAs known as miRNAs average 18 to 22 nucleotides and generally act as negative regulators in the cytoplasm by preventing translation or accelerating mRNA degradation by binding to the 3′-UTR of target genes [12]. Many miRNAs have been found to be substantially expressed in skeletal muscle and play an important role in muscle growth. For example, by targeting MyoD1, bta-miR-885 promotes proliferation while inhibiting differentiation in bovine skeletal muscle [13]. In addition, miR-133, miR-206, and miR-1 have been shown to be muscle-specific miRNAs with important regulatory functions in skeletal muscle growth and development [14,15,16]. Currently, miRNAs have been found to be involved in buffalo muscle development. Huang J et al. for the first time compared the miRNA composition of buffalo muscle and adipose tissue and identified miRNAs potentially related to buffalo intramuscular fat [17]. However, the potential regulatory mechanisms and roles of miRNAs in buffalo muscle development need to be further elucidated.

In summary, in this study, miRNAs related to the muscle development of buffaloes were screened via sequencing and combined through a bioinformatics analysis. We selected and investigated one of the new miRNAs, bbu-miR-493-5p, which is expressed at all stages of cell proliferation and differentiation with highly significant differences. It promotes buffalo myoblast proliferation and differentiation and is highly expressed in the muscular tissue of buffalo muscle. Our study lays the foundation for further exploring the miRNA regulatory mechanism of buffalo skeletal muscle development and provides a theoretical basis for finding molecular markers for further buffalo improvement and breeding.

## 2. Materials and Methods

### 2.1. Cell Sample Sources

In this study, three-month-old (*n* = 6) native Guangxi swamp buffalo calves from a local slaughterhouse in Nanning were used as the experimental subjects. After cleaning and sterilizing the samples with alcohol and normal saline, we collected longissimus dorsi muscle tissue, isolated the buffaloes’ primary myoblasts, and cultured the buffalo myoblasts at the proliferation stage (0 day) and at the differentiation stage (6 days). An miRNA sequencing analysis was performed on these samples.

### 2.2. Library Construction and sRNA Sequencing

The concentration of RNA samples was detected using a Qubit^®^ 3.0 Flurometer; the quality of RNA was assessed by means of an RNA Nano 6000 Assay in an Agilent 2100 Bioanalyzer. The small RNA sequencing (sRNA-seq) samples were designed to include three biological replicates each in the proliferative (BuffGM) and differentiation (BuffDM) phases, and each replicate was constructed with 10 μg of total RNA as the raw material for a small RNA sequencing library. The isolation of small RNAs 18–30 nt in length from the total RNA samples was achieved using PAGE gel at a 15% concentration and recovery via gel cutting. The small RNA was enriched using an ethanol precipitation. The library was constructed with a TruSeq Small RNA Sample Prep Kit (Illumina, Shanghai, China) according to the standard procedure, including the following steps: ligation of the 3′-end and 5′-end adapters; transcription in reverse and PCR amplification; and recovery of the target fragments (145–160 bp) by means of 6% PAGE gel cutting. The library concentrations were determined after library construction using Qubit 3.0 fluorescence quantification. Agilent 2100 was used to assess the library’s quality, and the StepOnePlus real-time fluorescence quantitative PCR machine was used to precisely measure the library’s effective concentration (the concentration was required to be >10 nM). Finally, the QC-qualified libraries were subjected to single-end 50 bp sequencing on the HiSeq 2500 platform, and the obtained sequence data were to be analyzed in a subsequent bioinformatics analysis.

### 2.3. Treatment of Raw Sequencing Data

In this study, the Illumina platform was used for single-end 50 bp sequencing, and each library obtained about 1 Gb of raw reads, which were strictly filtered to ensure data quality. The quality of reads was analyzed using the fastp (v0.12.6) software to identify and remove low-quality reads, and the filtering criteria included the following: eliminating the reads containing splice sequences; eliminating the reads with more than 10% N bases; eliminating the reads consisting of all A bases; and removing the reads that made up more than half of the total and had a base quality score of Q ≤ 20. High-quality clean reads were obtained by filtering the raw reads, and the clean reads were then mapped to the buffalo reference genome using the Bowtie program (GCA_003121395.1). The standard settings were as follows: no mismatch was allowed at the chromosome level; and one mismatch was allowed at the scaffold level. Finally, the mapping rate and the total number of mapped reads were tallied.

### 2.4. Identification and Analysis of miRNAs

The miRbase (v21) database (https://mirbase.org/; accessed on 28 April 2019) was used to determine known miRNAs, and miRdeep2 (v2.0.0.8) was used to predict new miRNAs. An analysis of variance was performed with DESeq (1.16.0), and the data were statistically standardized using the negative binomial distribution method. The misclassification rate was decreased using both the Hochberg and Benjamini approaches. *p* ≤ 0.05 and |log2(Fold_change)| ≥ 1 were used to filter for differentially expressed miRNAs.

### 2.5. GO Enrichment Analyses

In this study, a GO annotation analysis (https://geneontology.org/; accessed on 15 May 2019) was used to explore the potential functions of miRNA target genes. By functionally annotating genes in terms of biological processes, cellular structure, and molecular function, this method not only predicts the function of individual genes but also simplifies the complexity of the process.

### 2.6. Synthesis and Primer Design of miRNA Mimic

The miRNA mimic was synthesized by RiboBio, and the stem–loop primers and upstream and downstream primers were designed according to the principle of the miRNA stem–loop primer design. Table 1 lists all the primer sequences.

### 2.7. Synthesized cDNA and qRT-PCR Assays

From the cells treated with Trizol (Vazyme, Nanjing, China) for RNA extraction, 1 pg–1 μg of RNA was taken and added to 4× gDNA wiper Mix to remove the genome. Then, 4 μL of 5× HisScript III qRT SuperMix (Vazyme, Nanjing, China) and 0.5 μL of miRNA stem–loop primer were added and mixed, and reverse transcription was performed to generate cDNA. The enzyme used for quantitative detection was Cham QTM Universal SYBR^®^ qPCR Master Mix (Vazyme, Nanjing, China). Table 1 lists the primers used.

### 2.8. Buffalo Myoblast Isolation and Culture

The longest dorsal muscle of a three-month-old calf was taken and isolated into muscle cells. Skin, fascia, and other tissues were stripped from the longest dorsal muscle tissue of fetal bulls, and the muscle tissue was minced into the smallest possible pieces with scissors. The tissue was shaken with collagenase *I* (Solarbio, Beijing, China) for 1 h at 37 °C and then centrifuged at 1500× *g* to obtain the precipitate. Subsequently, trypsin was added for 20–25 min, and digestion was terminated with a DMEM medium containing 10% fetal bovine serum (FBS) (Gibco, Waltham, MA, USA). The cells were then filtered through 100 mesh and 70 mesh filters and centrifuged at 1500 rpm to obtain a cell precipitate. Finally, the cells were resuspended in a medium containing 20% FBS (Gibco, Waltham, MA, USA) and 1% penicillin-streptomycin (Gibco, Waltham, MA, USA). After 2 h of incubation in a 37 °C incubator, the cell cultures and supernatants were inoculated into cell culture plates at 37 °C and 5% CO_2_ for a subsequent passaging culture. When cell proliferation reached 80%, a differentiation medium containing 2% horse serum (Gibco, Waltham, MA, USA) was used instead of the growth medium to induce cell differentiation.

### 2.9. CCK-8 Analysis

Buffalo myoblasts were seeded into 96-well plates, cultured to a density of approximately 60%, and transfected with the ExFect transfection reagent (Vazyme, Nanjing, China) for 15–17 h, after which 10 μL of Cell Counting Kit-8 (CCK-8) working solution (Beyotime Biotechnology, Shanghai, China) was added, followed by a 3 h incubation period. The cell proliferation rates were obtained by measuring absorbance at 450 nm with a microplate reader (TECAN, Menedorf, Switzerland).

### 2.10. EdU Analysis

An EdU Cell Proliferation Kit (RiboBio, Guangzhou, China) was used to assess the proliferation of buffalo myoblasts. At 37 °C and 5% CO_2_, cells were introduced in 96-well plates and cultivated. Transfection was performed when the density reached 60% for 16–18 h. The cells were subsequently given an 8 h treatment with 10 μM of EdU reagent. After being fixed with PBS containing 4% paraformaldehyde for 20 min, the samples were treated with 2 mg/mL glycine for 10 min at room temperature. Then, they were incubated with Triton X-100 at a concentration of 0.5% for 20 min. Apollo^®^ staining was carried out for 30 min in accordance with the kit’s instructions. Afterwards, the samples were PBS-rinsed twice for 10 min. Finally, Hoechst33342 staining was performed for 30 min, washed with PBS, and photographed with an EVOS fluorescence microscope (Thermo, Waltham, MA, USA).

### 2.11. Western Blotting

Proteins were extracted with a RIPA (SolarBio, Beijing, China) buffer containing 1% PMSF (Beyotime Biotechnology, Shanghai, China). A BCA kit (Solarbio, Beijing, China) was used to measure protein concentration. The extracted proteins were separated by electrophoresis on polyacrylamide gels with a 10% SDS content and then transferred to polyvinylidene fluoride membranes using a semi-wet procedure. Antibodies against MyoD1 (Wanlei, Shenyang, China), MyHC, and β-actin (Abclonal, Wuhan, China) were incubated with the membranes at 4 °C overnight. The following day, the antibodies were incubated for 1 h at room temperature with the relevant HRP-conjugated secondary antibody. The membranes were washed with TBST and TBS, followed by incubation with the ECL Plus chemiluminescent agent (Solarbio, Beijing, China). Finally, the bands were imaged using a Chemidoc XRS+ system (Bio Rad, Irvine, CA, USA).

### 2.12. Immunofluorescence Staining (IF)

The buffalo myoblasts were fixed with paraformaldehyde for 20 min (4 °C). The cells were then rinsed with 0.5% Triton X-100 for 10 min, followed by incubation with 5% bovine serum albumin (BSA) on a shaker for one hour. The MyHC (Abclonal, Wuhan, China) antibody was incubated in the cells at 4 °C overnight. At the end of each of the above steps, the cells were rinsed three times with PBS for 5 min each time. After adding the fluorescently labeled secondary antibody (Abclonal, Wuhan, China), the sample was exposed to light-protecting conditions for 1 h while being incubated at room temperature. The sample was then rinsed for a minimum of two times with cold PBS. The nuclei were stained for 10 min with DAPI (2-(4-amidinphenyl)-6-indolecarba-midine dihydrochloride, cell signaling), which was then washed out with cold PBS. Finally, photographs were taken with an EVOS fluorescence microscope (Thermo, Waltham, MA, USA).

### 2.13. Fluorescence In Situ Hybridization (FISH)

The inoculated cells were grown to a 70% density on sterile slides before being treated with 4% paraformaldehyde for 20 min. The cells were treated with the prepared premix at 37 °C for 1 h. After being washed with PBS, the cells were treated overnight in a solution that also contained the bbu-miR-493-5p probe. The cells were incubated for 10 min without light after the addition of the DAPI solution. Afterwards, image capture with an EVOS fluorescence microscope took place (Thermo, Waltham, MA, USA).

### 2.14. Statistical Analysis

All the experiments were repeated at least three times, and data from a representative experiment were provided. All the data were expressed as the mean ± SEM of at least three biological replicates. The collected data were analyzed statistically using GraphPad (v6.0). A Student’s *t* test was used to evaluate the statistical significance of the difference. A *p*-value less than 0.05 was considered statistically significant, and the difference was indicated with an asterisk (*** *p* < 0.001; ** 0.001 < *p* < 0.01 and * 0.01< *p* < 0.05). This study’s raw sequencing data files can be found at the National Center for Biotechnology Information (NCBI) with the SRA ID PRJNA639027 (https://www.ncbi.nlm.nih.gov/bioproject/?term=PRJNA639027; accessed on 15 May 2020).

## 3. Result

### 3.1. Identification of miRNAs Associated with Buffalo Myoblast Development

In order to explore the potential biological role of miRNAs in skeletal muscle, we analyzed the expression of miRNAs during the proliferation (*n* = 3) and differentiation (*n* = 3) phases of buffalo myoblast. In the six buffalo myoblasts libraries established, between 22,246,677 and 27,37,804 and between 29,381,808 and 39,433,571 clean reads were obtained in the proliferation and differentiation stages. When comparing the sequences with the buffalo genome, 14,376,236~18,660,464 and 20,162,103~24,809,247 clean reads were localized in the genome, both with a comparison rate of more than 60% (Table 2). The number of known miRNA clean reads species was 5619~6228, and the total number of reads was 10,562,218~14,242,873 in the BuffGM library. The number of known miRNA clean reads species was 5633~6154, and the total number of reads was 12,950,856~15,533,679 in the BuffDM library (Table 2). Finally, the BuffGM and BuffDM libraries identified a total of 730 known miRNAs and 221 novel miRNAs (Appendix A). Among them, 59~80 novel miRNAs and 16,242~43,552 novel miRNA clean reads were obtained in the BuffGM library, while 40~52 novel miRNAs and 6433~43,547 novel miRNA clean reads were obtained in the BuffDM library (Table 2).

### 3.2. Screening of Differentially Expressed miRNAs

A total of 177 differentially expressed miRNAs were identified by the DEseq (1.16.0) software, of which 88 were significantly up-regulated, and 89 were significantly down-regulated (*p* < 0.05, Figure 1A–D, Appendix A). In order to deeply analyze the features of these miRNAs, we performed a hierarchical clustering analysis (Figure 1E–G). The findings demonstrated that the six small RNA sequencing libraries were essentially split into two groups based on the proliferation and differentiation phases, with comparable biological reproducibility within the two phases, following our clustering analysis. This suggests that the differentially expressed miRNAs identified in this study reflect the differences in miRNA transcripts in buffalo myoblasts during the proliferative and differentiation phases as well as the reliability of the identified miRNAs.

### 3.3. Candidate Target Gene Prediction and GO Analysis

To further investigate the biological functions of miRNAs, we anticipated the target genes of all significantly differentially expressed miRNAs using miRanda (v3.3a) and TargetScan (v3.1). The sum of 824 candidate target genes was identified. (Figure 2A and Appendix A). To determine the major biological functions of the target genes, a GO clustering enrichment analysis was performed (Figure 2B,C), which revealed that the candidate target genes of the significantly differentially expressed miRNAs were mainly involved in cellular processing, the bioregulatory metabolism, and the organization and biogenesis of cellular components. These were associated with molecular functions and transport functions related to cellular components enriched in organelles as well as binding and catalytic activities. Among them, Novel-214 is predicted to bind the target gene SELE, which is predominantly enriched in the calcium-binding-related pathway.

### 3.4. Identification of bbu-miR-493-5p as a Candidate miRNA

By analyzing the small RNA sequencing data of buffalo myoblasts during the proliferative and differentiation phases, we focused on the only novel miRNA, Novel-214, among the miRNAs expressed in all six sequencing libraries with significant differences in the proliferation and differentiation stages (Appendix A). The tissue expression profile showed that the expression of Novel-214 in buffalo skeletal muscle was significantly higher than that registered in other organs (heart, liver, spleen, and lung) (Figure 3A). Compared with the buffalo myoblast proliferation stage, the expression of Novel-214 was significantly higher in the differentiation stage (*p* < 0.001, Figure 3B), suggesting that this miRNA may play a role in the regulation of buffalo muscle development. Sequence conservation analyses revealed it to be identical to the miR-493-5p matrices of other species (human, goat, rat, etc.) (Table 3). Subsequently, we named it bbu-miR-493-5p for further study. Nucleoplasmic separation and fluorescence in situ hybridization (FISH) experiments showed that bbu-miR-493-5p was mainly located in the cytoplasm (Figure 3C), suggesting that bbu-miR-493-5p may be able to achieve gene silencing at the post-transcriptional level by binding to mRNAs in the cytoplasm, promoting mRNA degradation, or inhibiting translation. In addition, in order to investigate the role of bbu-miR-493-5p in the proliferation and differentiation of buffalo myoblasts, we successfully isolated buffalo myoblasts and performed Pax7 (marker of skeletal muscle satellite cells) and MyHC (marker of myoblast differentiation and maturation) staining (Figure 3C).

### 3.5. Effect of bbu-miR-493-5p on the Proliferation of Buffalo Myoblasts

We transfected a bbu-miR-493-5p mimic into buffalo myoblasts to greatly increase the expression of bbu-miR-493-5p in order to investigate the function of bbu-miR-493-5p in buffalo myoblasts proliferation (*p* < 0.01, Figure 4A). The effect of bbu-miR-493-5p on cell proliferation was measured using CCK-8, and the results showed that bbu-miR-493-5p significantly promoted cell viability (*p* < 0.05, Figure 4B). However, there was no discernible rise in EdU assay-positive cells (*p* > 0.05, Figure 4D and Appendix A). The results of the qPCR indicated that bbu-miR-493-5p significantly up-regulated the expression of CDK2, and the expression of PCNA was increased but not significant (Figure 4C). The above results indicated that bbu-miR-493-5p promoted the proliferation of buffalo myoblasts.

### 3.6. Effect of Buffalo Myoblast Differentiation by bbu-miR-493-5p

In order to investigate the regulatory role of bbu-miR-493-5p in the differentiation process of buffalo myoblasts, we transfected a bbu-miR-493-5p mimic into the cells. Then, after 6 days of inducing differentiation, qPCR was used to detect the expression of MyoD1, MyoG, and MyHC. The findings demonstrated that the overexpression of bbu-miR-493-5p greatly boosted the expression of three genes (*p* < 0.01, Figure 5A). Results from Western blotting revealed a propensity for MyoG (*p* > 0.05) to rise and a significantly increase in *MyoD1* (*p* < 0.05, Figure 5B,C). The results of IF showed that bbu-miR-493-5p significantly promotes the formation of myotubes in buffalo myoblasts (*p* < 0.05, Figure 5D–F). These results indicate that bbu-miR-493-5p promotes the differentiation of buffalo myoblasts.

## 4. Discussion

Muscle fiber characteristics and skeletal muscle mass are highly related to economic traits such as meat quality and yield in beef cattle [18]. The development of bovine skeletal muscle is related to gene expression, and miRNAs play an important role in regulating gene expression. Therefore, the in-depth study of miRNAs and their mechanism of action in regulating the development of bovine skeletal muscle will provide valuable information for the genetic breeding of cattle. Increasingly, studies have shown that miRNAs are involved in the regulation of muscle development. For example, bta-miR-365-3p targets the activin A receptor type I to reduce proliferation and increase differentiation in primary bovine myoblasts [18]. Cacchiarelli et al. found that myasthenia gravis was associated with altered miRNA expression in the mdx mouse model, in which miRNAs such as miR-1 and miR-133a were dependent on the regulation of the dystrophin pathway [19].

In this study, 177 differentially expressed miRNAs in the proliferation and differentiation phases of buffalo myoblasts were identified using sRNA-seq technology. We screened the differentially expressed miRNAs and identified a novel miRNA, bbu-miR-493-5p. Its expression is significantly up-regulated during the differentiation phase of buffalo myoblasts. The tissue expression profile analysis showed that the expression of bbu-miR-493-5p was significantly higher in buffalo skeletal muscle than in other organs = (heart, liver, spleen, lung), suggesting that it plays an important role in buffalo skeletal muscle development and is worth researching. miR-493-5p has been widely used as a tumor suppressor in various cancers, such as hepatocellular carcinoma [20,21], cervical cancer [22], renal cell carcinoma [23], gastric cancer [24], and so on. In recent years, miR-493-5p has also been found to be associated with skeletal muscle growth and development. For example, miR-493-5p promotes porcine myoblast proliferation and inhibits differentiation by targeting ANKRD17 [25]. Based on this, we inferred that it may play an important role in buffalo skeletal muscle development. Therefore, we explored the effect of bbu-miR-493-5p on buffalo myoblast proliferation and differentiation. In our work, we found that the overexpression of bbu-miR-493-5p significantly up-regulated the expression of the proliferation marker gene CDK2. The CCK8 and EdU assays showed that its proliferation efficiency was increased compared to the control group. This promotion of proliferation is similar to the results of miR-493-5p in a study of porcine myoblasts by Zhuang et al. [25]. During buffalo myoblast differentiation, we found that the overexpression of bbu-miR-493-5p significantly up-regulated the expression of the differentiation marker genes MyoD1, MyoG, and MyHC. The immunofluorescence results showed that the overexpression of bbu-miR-493-5p significantly promoted the fusion of myoblasts and myotube formation. These results indicate that bbu-miR-493-5p promotes buffalo myoblast differentiation. Interestingly, this promotion of differentiation is contrary to the results of miR-493-5p in porcine myoblasts, found in Zhuang et al.’s study [25]. Therefore, we infer that the function of miR-493-5p is not conserved in different species.

To further explore the regulatory mechanism of bbu-miR-493-5p, nucleoplasmic localization experiments revealed that bbu-miR-493-5p was mainly localized in the cytoplasm and partially expressed in the nucleus. Different localizations of miRNAs in cells play different functions, suggesting that bbu-miR-493-5p may play a regulatory role in buffalo myoblasts through differential expression in the cytoplasm and nucleus. In the cytoplasm, miRNAs mainly bind to the 3’-UTR of target genes to inhibit translation or promote mRNA degradation, thereby playing a negative regulatory role [12]. Predictive analysis of the bbu-miR-493-5p target gene revealed that it binds to SELE, which encodes the protein E-selectin. E-selectin is a cell adhesion molecule involved in the adhesion of leukocytes to vascular endothelial cells, and it promotes leukocyte infiltration into tissues [26]. When muscle injury occurs, inflammatory factors and the chemokines produced at the site of injury recruit leukocytes to accumulate at the site of injury and remove dead cells and necrotic tissue, which is essential for muscle regeneration. However, a moderate inflammatory response promotes muscle regeneration, whereas an excessive inflammatory response inhibits muscle regeneration [27]. Precise modulation of the extent of the inflammatory response is essential to ensure normal regenerative repair after muscle injury. Therefore, we hypothesized that bbu-miR-493-5p may regulate leukocytes infiltration into damaged muscle tissue by inhibiting the expression of SELE, thus playing a regulatory role in the inflammatory response and regenerative process after muscle injury. On the other hand, our enrichment pathway analysis showed that SELE is in a calcium ion-binding-related pathway, suggesting that its encoded protein, E-selectin, may function by binding to calcium ions. It has been found that calcium chelators can block SELE-mediated leukocyte adhesion through the inhibition of calcium-dependent binding [28]. This provides additional insight for us to further investigate the inflammatory response in bbu-miR-493-5p treatment of muscle injury.

On the other hand, bbu-miR-493-5p in the nucleus may bind to promoters, enhancers and other regulatory elements to affect the transcriptional expression of genes. There have been several studies targeting miRNAs in the nucleus. For example, nuclear-activating RNA (NamiRNA) localizes itself in the nucleus and activates proximal or distal genes by targeting enhancers [29]. Dongen Ju et al. found that miR-24-1 inhibited the Warburg effect in cancer cells by targeting the active enhancer in the nucleus to activate FBP1 transcription and slow down aerobic glycolysis, ultimately preventing the development of renal cell carcinoma [30]. Younger et al. designed miRNA mimics that specifically bind to target gene promoters and directly inhibit gene transcription [31]. Therefore, bbu-miR-493-5p may bind to promoters, enhancers, and other regulatory elements of proliferation- and differentiation-related genes in the nucleus, thereby influencing their expression, that, in turn, regulate the development of buffalo skeletal muscle.

## 5. Conclusions

In this study, we identified 177 miRNAs that were differentially expressed in buffalo myoblasts during the proliferation and differentiation phases. Further studies revealed that a novel miRNA, bbu-miR-493-5p, significantly increased the expression of marker genes for the proliferation and differentiation of buffalo myoblasts and promoted the fusion of myoblasts into myotubes. This study provides a new research basis for miRNA regulation of skeletal muscle development in buffaloes and provides new molecular targets for beef cattle breeding.

## Figures and Tables

**Figure 1 animals-14-00533-f001:**
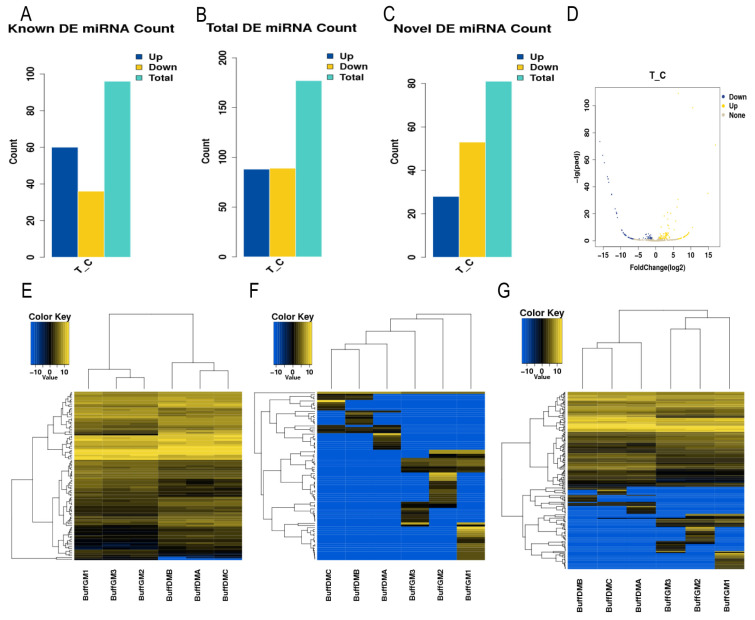
Differentially expressed miRNAs in buffalo myoblasts. (**A**–**C**) Statistical plots of the number of differentially expressed miRNAs (known miRNAs, unknown miRNAs, total miRNAs). (**D**) Volcano plots of the total differentially expressed miRNAs. (**E**–**G**) Hierarchical cluster analysis plots of the differentially expressed miRNAs (known miRNA, unknown miRNA, total miRNA) in buffalo myoblasts in the proliferative stage versus the differentiation stage.

**Figure 2 animals-14-00533-f002:**
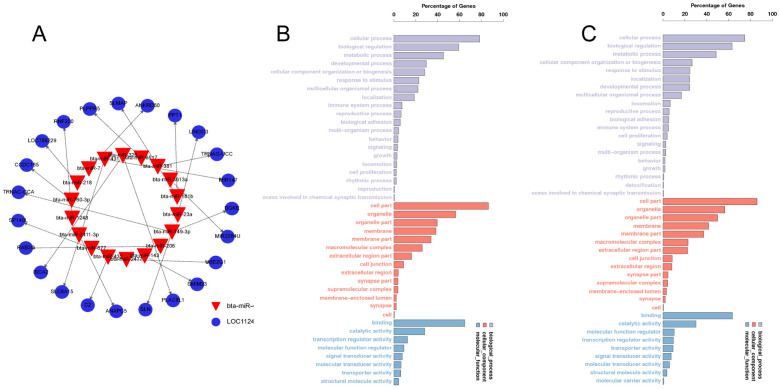
Candidate target gene prediction and GO analysis. (**A**) Network diagram of some candidate target genes. (**B**,**C**) Target gene GO clustering enrichment analysis diagrams.

**Figure 3 animals-14-00533-f003:**
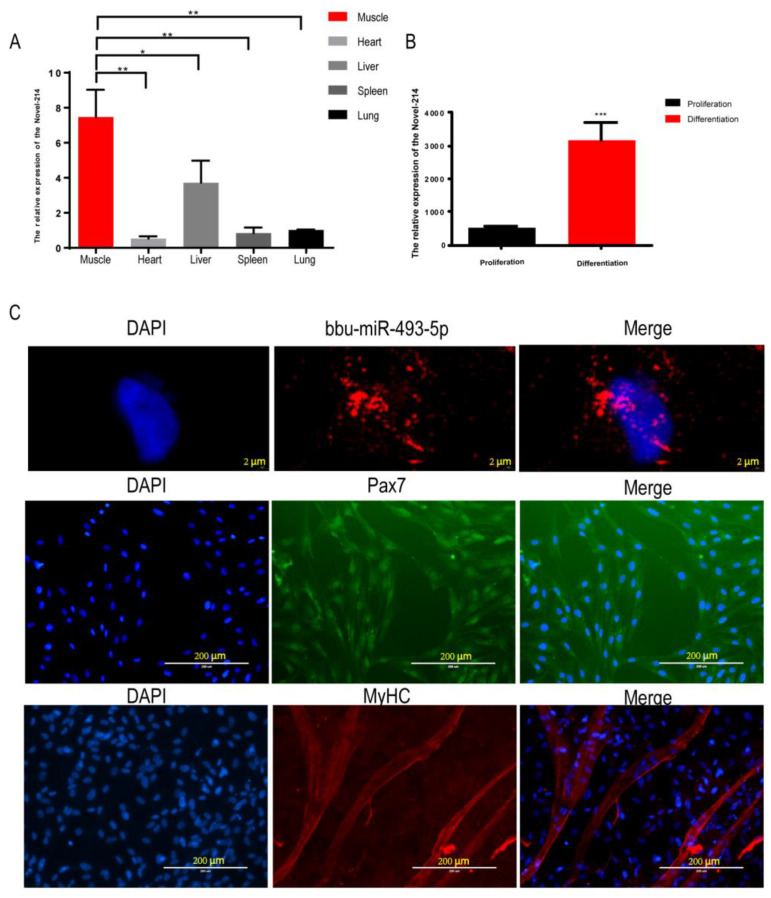
Identification of bbu-miR-493-5p as a candidate miRNA. (**A**) Tissue expression profile of Novel-214 in buffalo during fetal stage. (**B**) Expression of Novel-214 in buffalo myoblasts during proliferation and differentiation. (**C**) RNA-FISH detection of bbu-miR-493-5p cellular localization; Pax7 immunofluorescence staining graph; and MyHC immunofluorescence staining graph. The data represent the mean ± SEM of at least three independent experiments. *** *p* < 0.001; ** 0.001 < *p* < 0.01; and * 0.01< *p* < 0.05.

**Figure 4 animals-14-00533-f004:**
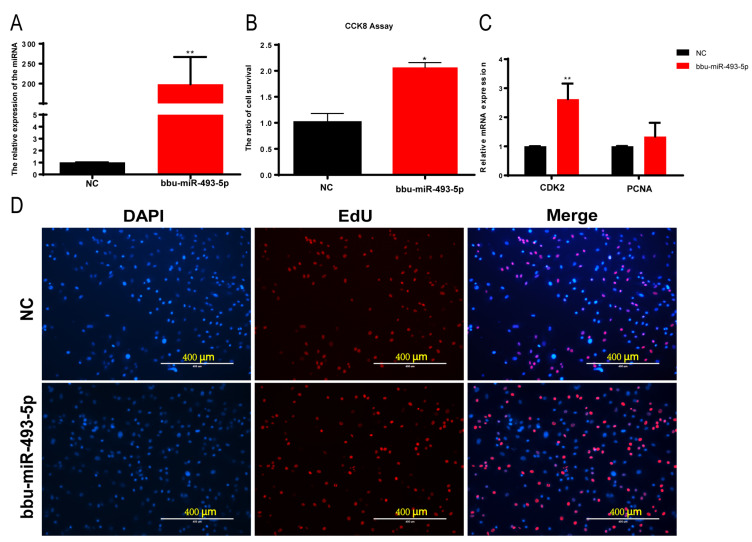
CCK-8, EdU, and qPCR assays to detect the effect of overexpression of bbu-miR-493-5p on the proliferation of buffalo myoblasts. (**A**) bbu-miR-493-5p overexpression efficiency assay. (**B**) CCK-8 assay to detect the effect of overexpression of bbu-miR-493-5p on the proliferation of adult myoblasts. (**C**) qPCR assay to detect the expression of proliferation marker genes in buffalo myoblasts after overexpression of bbu-miR-493-5p. (**D**) EdU fluorescence assay to detect the effect of overexpression of bbu-miR-493-5p on the proliferation of adult myoblasts. The data represent the mean ± SEM of at least three independent experiments. ** 0.001 < *p* < 0.01; and * 0.01 < *p* < 0.05.

**Figure 5 animals-14-00533-f005:**
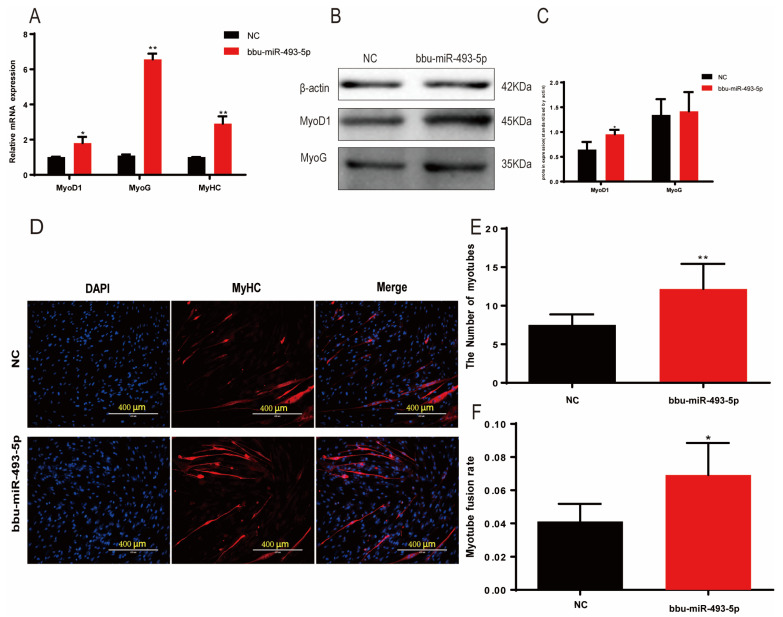
bbu-miR-493-5p promotes the differentiation of buffalo myoblasts. (**A**) qPCR to detect the expression of differentiation marker genes in buffalo myoblasts after overexpression of bbu-miR-493-5p. (**B**,**C**) WB to detect the effect of overexpression of bbu-miR-493-5p on buffalo myoblasts differentiation. (**D**) Immunofluorescence detection of MyHC expression. (**E**) Statistics of changes in the number of myotubes. (**F**) Statistics of changes in myoblast fusion rate. The data represent the mean ± SEM of at least three independent experiments. ** 0.001 < *p* < 0.01; and * 0.01 < *p* < 0.05.

**Table 1 animals-14-00533-t001:** Primer sequence list.

Gene or miRNA	Primer Sequences (5′-3′)
PCNA	F: TCCAGAACAAGAGTATAGC
	R: TACAACAGCATCTCCAAT
CDK2	F: TTTGCTGAGATGGTGACCCG
	R: TAACTCCTGGCCAAACCACC
MyoD1	F: CCCAAAGATTGCGCTTAAGTG
	R: GTTCCTTCGCCTCTCCTACCT
MyoG	F: CAAATCCACTCCCTGAAA
	R: GCATAGGAAGAGATGAACA
MyHC	F: TGCTCATCTCACCAAGTTCC
	R: CACTCTTCACTCTCATGGACC
β-actin	F: CATCCTGACCCTCAAGTA
	R: CTCGTTGTAGAAGGTGTG
U6	F: CGCTTCGGCAGCACATATAC
	R: AAATATGGAACGCTTCACGA
bbu-miR-493-5p	F: GCGCTTGTACATGGTAGGCT
	R: GTGCAGGGTCCGAGGT
bbu-miR-493-5p	RT:GTCGTATCCAGTGCAGGGTCCGAGGTATTCGCACTGGATACGACAATGAA

**Table 2 animals-14-00533-t002:** Comparative statistics of clean readings and reference sequences.

Sample	BuffDMA	BuffDMB	BuffDMC	BuffGM1	BuffGM2	BuffGM3
Total Reads	39,433,571	31,105,328	29,381,808	22,246,677	27,537,804	27,144,870
Perfect Match Reads	24,809,247	20,963,513	20,162,103	14,376,236	17,860,587	18,660,464
Perfect Match Rate (%)	62.91	67.4	68.62	64.62	64.86	68.74
Not Match Rate (%)	37.09	32.6	31.38	35.38	35.14	31.26
Total Clean Reads Mature	15,533,679	14,335,037	12,950,856	10,562,218	12,981,215	14,242,873
Known miRNA Number	597	593	587	620	609	592
Novel miRNA Number	40	52	51	80	66	59
Total Clean Reads Novel	16,382	6433	43,547	43,552	36,879	16,242

**Table 3 animals-14-00533-t003:** Sequence comparison table.

miRNA	Sequence	Comparison Rate
bbu-miR-493-5p	UUGUACAUGGUAGGCUUUCAUU	100%
hsa-miR-493-5p	UUGUACAUGGUAGGCUUUCAUU	100%
chi-miR-493-5p	UUGUACAUGGUAGGCUUUCAUU	100%
rno-miR-493-5p	UUGUACAUGGUAGGCUUUCAUU	100%

## Data Availability

Data are contained within the article.

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
