# Peer review of "Buffalo bbu-miR-493-5p Promotes Myoblast Proliferation and Differentiation"

_animals, 2024, doi:10.3390/ani14040533_

Round 1

Reviewer 1 Report

Comments and Suggestions for Authors

Review:

The manuscript sent for review covers a very important topic; however, the authors were not safe from minor errors. The manuscript can be revised and improved and accepted after minor corrections have been made:

1)    The introduction section describes the regulatory mechanisms of muscle development in detail, but lacks any introductory information on Buffalo breeding and production, which the authors only mention in the last sentence of the introduction.

2)    CRITICAL NOTE: The description of the statistics fully needs to be corrected. How statistical errors can be expressed as mean and sem. Data are described as mean and sem. Also, it was not the p-values that were analysed with the Student's t-test. This should be corrected. What statistical program was used to perform the analyses? In order to use the Student's t-test, conditions must be met (e.g. normal distribution). Describe which tests met these conditions. Did all data meet these conditions, if not which non-parametric tests were used.

3)    Line 215: Results - Why were only n=3 used to assess miRNA expression during the proliferation and differentiation phase? It would be useful to add information here as to why these 3 were used.

4)    CRITICAL NOTE: Results: Figures should be improved: please note e.g. Fig. 4 C: CDK2 - the relevance on the red bar practically merges with the error bars or e.g. the microscopic images in Figs. 3, 4, 5 - the scale applied to the images is invisible even at multiple magnification, so the scale information can be added to the figure description which follows.

5)    CRITICAL NOTE: Figure 3A: Was bbu-miR493-5p expression compared in different organs? It is not clear - if so, from the graph the error bars show significant differences that are not highlighted. If they are not - the graph should be corrected. This is very unclear, this part should also be completed in the Results text.

6)    Line 245: The description of figure 1 is in bold in full, while the others are not - it would be appropriate to align this notation with all figures.

7)    CRITICAL NOTE: The discussion is written quite modestly - where it is the most important chapter of the manuscript and can very much 'advertise' the work. Yes some mechanisms are described in the discussion and supported by a very sparse literature. However, there is a definite lack of reference to papers that are already in the literature and comparison against their research results.

The manuscript needs a thorough re-review and many changes, after which it should be re-evaluated.

Author Response

Dear Reviewer,

Thank you for your correspondence and for the valuable comments provided by the reviewers regarding our manuscript titled “Buffalo bbu-miR-493-5p promotes myoblast proliferation and differentiation(animals-2755539). We sincerely appreciate the careful examination of our work and have thoroughly revised the comments raised by the reviewers. The revised sections of the manuscript have been appropriately marked for easy identification. The main corrections made in the manuscript, along with our responses to the reviewers' comments, are outlined below:

The manuscript sent for review covers a very important topic; however, the authors were not safe from minor errors. The manuscript can be revised and improved and accepted after minor corrections have been made:

1.The introduction section describes the regulatory mechanisms of muscle development in detail, but lacks any introductory information on Buffalo breeding and production, which the authors only mention in the last sentence of the introduction.

Response: We would like to express our sincere gratitude to the reviewers for their suggestions, and we have added a description of buffalo breeding and production in lines 35-42 of the manuscript as follows: Buffalo is an indispensable livestock resource in developing countries, originally used for agriculture, and now it has become one of the meat sources for human consumption[9]. Existing studies show that buffalo meat quality is no different from ordinary beef in appearance, and has great meat development value[12]. Therefore, the development and use of molecular technology to breed high-quality buffalo with fast growth, large meat yield and high quality can not only effectively improve buffalo varieties and protect buffalo germplasm resources, but also improve the quality and meat yield of buffalo beef and solve the problems of beef supply.

2.CRITICAL NOTE: The description of the statistics fully needs to be corrected. How statistical errors can be expressed as mean and sem. Data are described as mean and sem. Also, it was not the p-values that were analysed with the Student's t-test. This should be corrected. What statistical program was used to perform the analyses? In order to use the Student's t-test, conditions must be met (e.g. normal distribution). Describe which tests met these conditions. Did all data meet these conditions, if not which non-parametric tests were used.

Response: Thanks to the reviewers for your careful review and suggestions on the statistical methods of the article. According to your suggestions, we have carried out the following statistical analysis and method description improvement in the revised draft: All experiments were repeated at least three times and data from a representative experiment were provided. All data were expressed as mean ±SEM of at least 3 biological replicates. The collected data were analyzed statistically by GraphPad (v6.0). Student's t test was used to evaluate the statistical significance of the difference. A P value less than 0.05 was considered statistically significant, and the difference was indicated by an asterisk (*p <0.05; * p < 0.01; * * p < 0.001; * * * p < 0.0001).

3.Line 215: Results - Why were only n=3 used to assess miRNA expression during the proliferation and differentiation phase? It would be useful to add information here as to why these 3 were used.

Response: Thank you for question. Due to the limitation of research conditions, it is difficult to obtain a large sample size of fetal buffalo immediately in this study. Our fetal baffalo were all from pregnant female buffaloes mistakenly killed in slaughterhouses, and the number of fetal buffalo was very limited. However, despite the limited sample size, we still guaranteed three independent repeats of the experiment to improve the robustness and reliability of the results.

4.CRITICAL NOTE: Results: Figures should be improved: please note e.g. Fig. 4 C: CDK2 - the relevance on the red bar practically merges with the error bars or e.g. the microscopic images in Figs. 3, 4, 5 - the scale applied to the images is invisible even at multiple magnification, so the scale information can be added to the figure description which follows.

Response: Thanks to the reviewers for your suggestions to improve the manuscript. We have modified Fig. 4C and re-added the scale information for the images in manuscript.

5.CRITICAL NOTE: Figure 3A: Was bbu-miR493-5p expression compared in different organs? It is not clear - if so, from the graph the error bars show significant differences that are not highlighted. If they are not - the graph should be corrected. This is very unclear, this part should also be completed in the Results text.

Response: Thank you for pointing out the shortcomings. We have revised the original drawing, added the difference significance mark, and re-changed the description of significance in the manuscript, specifically in the manuscript 275-283 lines. The contents are as follows: By analyzing the small RNA sequencing data of buffalo myoblasts during the proliferative and differentiation phases, we focused on the only novel miRNA-Novel-214 among the miRNAs expressed in all six sequencing libraries with significant differences in proliferation and differentiation stages (Supplementary Table S4). Tissue expression profile showed that the expression of Novel-214 in buffalo skeletal muscle was significantly higher than that in other organs (heart, liver, spleen and lung) (Figure 3A). Compared with buffalo myoblast proliferation stage, the expression of Novel-214 was significantly higher in differentiation stage (p < 0.001, Figure 3B), suggesting that this miRNA may play a role in the regulation of buffalo muscle development.

6.Line 245: The description of figure 1 is in bold in full, while the others are not - it would be appropriate to align this notation with all figures.

Response: Thank you for your careful advice. We have made changes to the font and scrutinized the rest of the manuscript.

7.CRITICAL NOTE: The discussion is written quite modestly - where it is the most important chapter of the manuscript and can very much 'advertise' the work. Yes some mechanisms are described in the discussion and supported by a very sparse literature. However, there is a definite lack of reference to manuscripts that are already in the literature and comparison against their research results.

Response: Thank you for your careful comments. We would like to express our sincere thanks to the reviewers for your suggestions. We supplemented and modified the discussion, added some references, and compared and discussed our research content with the literature, specifically in lines 350 to 375 of the manuscript, which is as follows: In this study, 177 differentially expressed miRNAs in the proliferation and differ-entiation phases of buffalo myoblasts were identified using sRNA-seq technology. We screened the differentially expressed miRNAs and identified a novel miRNA, bbu-miR-493-5p. Its expression is significantly up-regulated during the differentiation phase of buffalo myoblasts. Tissue expression profile analysis showed that the expres-sion of bbu-miR-493-5p was higher in buffalo skeletal muscle than in other organs sig-nificantly (heart, liver, spleen, lung), suggesting it plays an important role in buffalo skeletal muscle development and is worth researching. The miR-493-5p has been widely used as a tumor suppressor in various cancers, such as hepatocellular carcino-ma[24; 27], cervical cancer[25], renal cell carcinoma[29], gastric cancer[15] and so on. In recent years, miR-493-5p has also been found to be associated with skeletal muscle growth and development. For example, miR-493-5p promotes porcine myoblast prolif-eration and inhibits differentiation by targeting ANKRD17[31]. Based on this, we in-ferred that it may play an important role in buffalo skeletal muscle development. Therefore, we explored the effect of bbu-miR-493-5p on buffalo myoblast proliferation and differentiation. In our work, we found that overexpression of bbu-miR-493-5p sig-nificantly upregulated the expression of the proliferation marker gene CDK2. CCK8 and EdU assays showed that its proliferation efficiency was increased compared to the control group. This promotion of proliferation is similar to the results of miR-493-5p in porcine myoblasts by Zhuang et al[31]. During buffalo myoblast differentiation, we found that overexpression of bbu-miR-493-5p significantly upregulated the expression of differentiation marker genes MyoD1, MyoG and MyHC. Immunofluorescence re-sults showed that overexpression of bbu-miR-493-5p significantly promoted fusion of myoblasts and myotube formation. These results indicate that bbu-miR-493-5p pro-motes buffalo myoblast differentiation. Interestingly, this promotion of differentiation is contrary to the results of miR-493-5p in porcine myoblasts by Zhuang et al[31]. Therefore, we infer that the function of miR-493-5p is not conserved in different species.

Reviewer 2 Report

Comments and Suggestions for Authors

ARTICLE REVIEW

Introduction

MicroRNAs (miRNAs) do not encode proteins, but they play a huge role in many biological processes, including the development and regeneration of skeletal muscles. miRNAs regulate the processes that take place during muscle reconstruction, from keeping satellite cells at rest, to their activation, proliferation and differentiation into myoblasts, to the fusion of these cells leading to the formation of miotubes. Many mRNAs have been found to be expressed in skeletal muscle and to play an important role in muscle growth. In this paper we investigated the influence of miRNA expression on the development and differentiation of myoblasts in Chinese buffalo.

Specific comments:

Title:

The title of the article is adequate to the presented research results.

Abstract: could be supplemented with the most important numerical results

Introduction:

The authors of the article adequately presented the research problem, using the appropriate literature

Material and Methods:

The research material and the analytical and statistical methods used were appropriate to the scope of this study. A more precise sampling procedure needs to be completed. Tell me what muscles the samples came from, how many calves they came from and what sex they were.

Results:

The results were collected in the form of 5 graphs and 3 tables with a description.

An important aspect would be to check the aging rate of fibroblasts undergoing cryopreservation.

Discussion:

The discussion chapter requires an extension and deeper discussion against the background of the scientific literature in this field and should refer to all the results obtained from the study.

Final remark:

This review paper addresses a very important scientific problem and provides information on the role of mRNA in the expression of genes associated with differentiation of skeletal muscles in buffalo.

Author Response

Dear Reviewer,

Thank you for your correspondence and for the valuable comments provided by the reviewers regarding our manuscript titled “Buffalo bbu-miR-493-5p promotes myoblast proliferation and differentiation” (animals-2755539). We sincerely appreciate the careful examination of our work and have thoroughly revised the comments raised by the reviewers. The revised sections of the manuscript have been appropriately marked for easy identification. The main corrections made in the manuscript, along with our responses to the reviewers' comments, are outlined below:

Introduction

MicroRNAs (miRNAs) do not encode proteins, but they play a huge role in many biological processes, including the development and regeneration of skeletal muscles. miRNAs regulate the processes that take place during muscle reconstruction, from keeping satellite cells at rest, to their activation, proliferation and differentiation into myoblasts, to the fusion of these cells leading to the formation of miotubes. Many mRNAs have been found to be expressed in skeletal muscle and to play an important role in muscle growth. In this paper we investigated the influence of miRNA expression on the development and differentiation of myoblasts in Chinese buffalo.

Specific comments:

1.Title:

The title of the article is adequate to the presented research results.

2.Abstract: could be supplemented with the most important numerical results.

Response: Many thanks to the reviewers for your valuable suggestions. At your reminder, We have carefully reviewed the summary again and confirmed that the most important data have been included. At present, I think the abstract has fully reflected our research results and provided the necessary numerical results.

3.Introduction:

The authors of the article adequately presented the research problem, using the appropriate literature.

4.Material and Methods:

The research material and the analytical and statistical methods used were appropriate to the scope of this study. A more precise sampling procedure needs to be completed. Tell me what muscles the samples came from, how many calves they came from and what sex they were.

Response: Thanks to the reviewer's suggestion, we revised the sample source part of the research method to make it more accurate, specifically in lines 84-89 of the manuscript, and modified the content as follows: In this study, 3-month-old (n = 6) native Guangxi swamp buffalo calves from a local slaughterhouse in Nanning were used as experimental subjects. After cleaning and sterilizing the samples with alcohol and normal saline, we collected longissimus dorsi muscle tissue, isolated buffalo primary myoblasts, and cultured buffalo myoblasts at proliferation stage (0 day) and differentiation stage (6 days). miRNA sequencing analysis was performed on these samples.

It is also worth noting that the sex of the six fetuses was random because the fetuses were so young that it was difficult to identify their sex.

5.Results:

The results were collected in the form of 5 graphs and 3 tables with a description.

An important aspect would be to check the aging rate of fibroblasts undergoing cryopreservation.

Response: This is a good question. You mentioned that checking the rate of aging of cryopreserved cells is an important aspect. The speed of cell aging directly affects the reliability of experimental results. In our study, the cells were frozen in p0 generation, and after thawing and recovery, the cell vitality was still strong, and it still had the ability to differentiate and fuse into muscle tubes. On the other hand, the cells we tested were all from the first three generations, so in this experiment we did not examine the effect of freezing on cell aging. Your valuable advice has given us good inspiration, and we will add details on the detection of frozen cell aging rate in future research plans. This will make our results more reliable and complete. Thank you again for your careful review and professional advice.

6.Discussion:

The discussion chapter requires an extension and deeper discussion against the background of the scientific literature in this field and should refer to all the results obtained from the study.

Response: Thank you for your suggestions. We have revised and supplemented the discussion to make it more complete. The details are in lines 339-375 of the manuscript.

Reviewer 3 Report

Comments and Suggestions for Authors

In this study, the impact of miRNAs on muscle development regulation is explored, with a focus on buffalo myoblasts. However, I have a few concerns/suggestion:

1. The percentage of reads match is low. should be at least 75% to be significant.

2. In fig 1D, is it total DE or novel DE?

3. In fig 1F, there is a huge variation in cluster analysis of unknown miRNA for both the phases. Authors should comment on this and what are the genes that display variation?

4. In line 262, the bbu-mir-493 should be denoted as Novel-214 because the reasoning for bbu-miR-493-5p is given later on.

5. Authors should rephrase thid para to match the figures. It is very misleading. bbu-miR-493-5p is highly expressed in differentiating myoblast compared to proliferative. This must be stated explicitly.

6. Fig 3c top panel merge fig is flipped. Should be corrected.

7. Lines 278-282: The miRNA could be present inside the nucleus to degrade mRNA as is made. I believe it is not necessary to give reasoning here unless there is any evidence in literature to support the reasoning.

8. Why was Pax7 and MyHC staining done? WHat do the results suggest? Please elaborate the text for broad audience. ALos, the images should be of same magnification. It is hard to see at low magnification. Pax7 signal looks low while, myHC shows background. I would suggest authors to change the imgaes with same magnification as used for FISH.

9. Overexpressing miR-493 could result in off-target effects. How would you address that?

10. CCK8 and EdU are showing different result.   How many times was EdU repeated? Imaging shows a clear difference though. 

11. In 3B, the miR-493 levels are low in proliferating myoblast, suggesting a negative role of bbu-mir-493 in myoblast proliferation. An increased proliferation upon mir-493 overexpression seems contradictory.

12. MyHC and DAPI images look like different magnifications. In merge, one myoblast seem to have more than one nucleus. I would suggest authors to provide a magnified and beeter image with clear scale bars.

ChatGPT

Comments on the Quality of English Language

Need to be improved

Author Response

Dear Reviewer,

Thank you for your correspondence and for the valuable comments provided by the reviewers regarding our manuscript titled “Buffalo bbu-miR-493-5p promotes myoblast proliferation and differentiation(animals-2755539). We sincerely appreciate the careful examination of our work and have thoroughly revised the comments raised by the reviewers. The revised sections of the manuscript have been appropriately marked for easy identification. The main corrections made in the manuscript, along with our responses to the reviewers' comments, are outlined below:

In this study, the impact of miRNAs on muscle development regulation is explored, with a focus on buffalo myoblasts. However, I have a few concerns/suggestion:

  1. The percentage of reads match is low. should be at least 75% to be significant.

Response: Thank you for the reviewer's attention to the quality of our miRNA sequencing data. The main reason for the 60-70% alignment rate of our miRNA sequencing data is that the clean reads were aligned to an imperfect reference genome.  When we collected and sequenced the samples, there was no specific reference genome available for swamp buffalo. Therefore, we used the Murrah buffalo genome as the reference, which led to the final perfect match rate being only 60-70%. We understand that the low alignment rate could affect the reliability of the results. To ensure the main conclusions remain valid, we verified the key results through multiple means, including replicate experiments and validation using other techniques. We found the main conclusions were not affected.

  1. 2. In fig 1D, is it total DE or novel DE?

Response: Thanks for the reviewer's questions. FIG. 1D refers to miRNAs with total differential expression, which was not clearly pointed out in our drawing notes. We are really sorry that we have modified this image annotation and carefully checked other image annotations.

  1. 3. In fig 1F, there is a huge variation in cluster analysis of unknown miRNA for both the phases. Authors should comment on this and what are the genes that display variation?

Response: We thank the reviewers for your keen insight and valuable comments. The cluster analysis of this graph does show that there are certain individual variations of unknown miRNAs. Due to the tissue specificity and timing of miRNAs, there may still be some sequences that are consistent with the characteristics of miRNAs but have not been discovered so far, which we consider to be unknown miRNAs. The expression of these unknown miRNAs may be different in different tissues or at different periods of the same tissue. Therefore, we speculated that the variation of the cluster analysis of unknown miRNAs in the two stages may be related to slight differences in physiological states among the sample individuals. We re-checked the cluster analysis process and confirmed that the method and parameter setting were reasonable. In addition, the unknown miRNA we are concerned about, Novel-214, has stable and high expression in each sample, so we confirm that Novel-214 is not the source of mutation, which will not affect our subsequent research results.

  1. In line 262, the bbu-mir-493 should be denoted as Novel-214 because the reasoning for bbu-miR-493-5p is given later on.

Response: Thanks to the reviewer for pointing out our mistake, we have changed the word bbu-miR-493-5p to Novel-214 on line 269 of the manuscript

  1. Authors should rephrase thid para to match the figures. It is very misleading. bbu-miR-493-5p is highly expressed in differentiating myoblast compared to proliferative. This must be stated explicitly.

Response: Thanks to the reviewers for their valuable comments, we strongly agree with you and have reworked the original text to clarify this fact. Details are in lines 280-282 of the manuscript.

  1. Fig 3c top panel merge fig is flipped. Should be corrected.

Response: We apologize for this obvious error and thank the reviewer for pointing it out. We have carefully checked other images to avoid such errors.

  1. 7. Lines 278-282: The miRNA could be present inside the nucleus to degrade mRNA as is made. I believe it is not necessary to give reasoning here unless there is any evidence in literature to support the reasoning.

Response: Thanks for the reviewer's valuable advice. After the reviewer's reminder, we found that the description here is indeed superfluous for the whole article, and we have deleted the description here.

  1. Why was Pax7 and MyHC staining done? What do the results suggest? Please elaborate the text for broad audience. ALos, the images should be of same magnification. It is hard to see at low magnification. Pax7 signal looks low while, myHC shows background. I would suggest authors to change the imgaes with same magnification as used for FISH.

Response: Thank the reviewers for their questions and suggestions. The purpose of Pax7 staining is to identify whether the cells we isolated are buffalo primary myoblasts, because it has been reported in the literature that Pax7 gene is a marker of skeletal muscle satellite cells (https://dx.doi.org/10.1007/s00590-019-02410-w). In order to ensure that the cells used in our subsequent experiments are buffalo skeletal muscle cells, the identification of pax7 is very necessary. In addition, the purpose of MyHC staining is to identify whether the isolated cells have differentiation ability, because it has been reported in literature that MyHC gene is a marker gene for the differentiation and maturation of myoblasts into muscle fibers (https://onlinelibrary.wiley.com/doi/10.1111/febs.14502) and our subsequent experimental design includes exploring the effect of bbu-miR-493-5p on the differentiation ability of buffalo myoblasts. Therefore, it is necessary to ensure that the isolated cells are capable of differentiation. In order for readers to clearly understand the purpose of our coloring, we have revised and supplemented the article, detailed in lines 295-300 of the manuscript. Thanks again to the reviewers for their questions and suggestions, which further improved the clarity of our article. In addition, we re-adjusted the clarity of the pictures and replaced them with images with higher magnification.

  1. Overexpressing miR-493 could result in off-target effects. How would you address that?

Response: Thank you reviewers for raising this very important question. The reviewer's concerns are very legitimate. In our experiment, we synthesized miR-493 mimics to ensure that our overexpression was successful and functional. At the same time, we also established a control group transfected with NC, and the phenotypic changes observed between the two groups supported the regulatory effect of miR-493 with certain specificity. The off-target effect you mentioned deserves further consideration. In future experiments, we will consider using plasmid-mediated miRNA overexpression, cloning the gene sequence encoding miRNA onto the plasmid vector, and then transfecting the recombinant plasmid into the cell, which may reduce the off-target effect.

  1. 10. CCK8 and EdU are showing different result. How many times was EdU repeated? Imaging shows a clear difference though.

Response: Thank you reviewer for a very good question. In the EDU experiment, the control group and the experimental group carried out the biological repetition of 5-8 holes respectively, and 5 photos were taken under different field of view of each hole. The counting and statistics were carried out by imagej software. Although the results after analysis were not significant, there was a trend of promoting proliferation, which was consistent with the trend of CCK8 results. We carefully re-examined the experimental procedure, and found that the inconsistency may be due to the lower sensitivity of the EdU method itself in detecting proliferating cells. Although immunofluorescence imaging showed some differences, quantitative analysis is needed to draw statistical differences. We agree with the reviewer's opinion. In the future, we will use more sensitive and reliable methods such as MTT, cell counting, etc. to validate and provide more accurate and reliable data.

  1. In 3B, the miR-493 levels are low in proliferating myoblast, suggesting a negative role of bbu-mir-493 in myoblast proliferation. An increased proliferation upon mir-493 overexpression seems contradictory.

Response: Thank the reviewers for their suggestions. We note that bbu-miR-493-5p does show different expression patterns and biological functions in different cell states. The level of bbu-miR-493-5p may be subject to complex regulation under normal physiological conditions, including regulation of other miRNAs, transcription factors, and cell signaling pathways. The low level of bbu-miR-493-5p in proliferation myoblasts may be due to the maintenance of normal proliferative state, and we introduced artificial intervention through overexpression of bbu-miR-493-5p, thus altering this balance. The role of miRNA in cells may involve multiple pathways and targets. Under our experimental conditions, bbu-miR-493-5p may lead to increased proliferation by regulating target genes or pathways related to proliferation.

  1. MyHC and DAPI images look like different magnifications. In merge, one myoblast seem to have more than one nucleus. I would suggest authors to provide a magnified and beeter image with clear scale bars.

Response: We thank the reviewers for their questions and suggestions. Due to our mistake, the image and the image scale were not clear enough to cause the reviewer's misunderstanding, for which we replaced the image with a higher magnification, re-added the scale to the image. In addition, MyHC and DAPI images were acquired at the same magnification of magnification. The reason why multiple nuclei were observed in a single cell is because the cell is at a mature stage of differentiation at this time, and the development of myoblasts to myotubes undergoes the process of fusion of multiple cells into a single myotube, which is the hallmark of myotube formation.

Round 2

Reviewer 1 Report

Comments and Suggestions for Authors

Thank you for completing and improving the manuscript according to the suggestions offered. Therefore, I accept the changes applied. 

Author Response

Dear Reviewer,

Thank the reviewer for accepting the changes we made to the manuscript “Buffalo bbu-miR-493-5p promotes myoblast proliferation and differentiation(animals-2755539). We are very pleased that these changes meet the review requirements. Your approval means a lot to us. We will continue my efforts to further improve the quality of our research in future work. Thank you again for your recognition and support of our thesis. We sincerely wish you success in your work and good health.

Reviewer 3 Report

Comments and Suggestions for Authors

I am mostly satisfied by the authors' comment on my review. However, to avoid confusion, I would suggest moving 4E to supplementary.

Other than this, I believe the edited manuscript is suitable for publication.

Comments on the Quality of English Language

NA

Author Response

Dear Reviewer,

Many thanks to the reviewers for your positive comments on our revised manuscript “Buffalo bbu-miR-493-5p promotes myoblast proliferation and differentiation(animals-2755539). We sincerely thank the reviewers for your careful review of the revised manuscript and suggestions. The main changes we made to the manuscript, and our responses to reviewer comments, are outlined below:

  1. I am mostly satisfied by the authors' comment on my review. However, to avoid confusion, I would suggest moving 4E to supplementary. Other than this, I believe the edited manuscript is suitable for publication.

Response: We are very grateful to the reviewers for their recognition of the revised manuscript. We would be happy to accept your suggestion to move Figure 4E from the manuscript to the Supplementary material and add "Supplementary Figure S1" on line 308 of the manuscript. We double-checked the manuscript to make sure that no other changes had been missed.